# Ecovillagers' Assessment of Sustainability: Differing Perceptions of Technology as a Differing Account of Modernism

Zach Rubin

Department of Government, Criminology, and Sociology, Lander University, Greenwood, SC 29649, USA; zrubin@lander.edu

**Abstract:** There is great debate over how to measure progress towards ecological sustainability, and a number of approaches deployed by various groups to try and achieve it. One of these is the "ecovillage," a form of communal living—the primary purpose of which is to enable a reduction in consumption through the collectivization of resources. This paper presents a case study of an ecovillage named Dancing Rabbit, which stands as an example counter to mainstream discourses on progress through the relatively successful approach to ecological sustainability. In the dwellings they have built, their resource consumption patterns, and the diverse attitudes towards technology use that they express, the ecovillagers in this study demonstrate narratives of progress which center an ecological motive over one of profitable innovation. Rather than rely on modernist assumptions of engineering our way to sustainable living through advanced technology or embrace primitivist notions of rejecting it all together, the case study of Dancing Rabbit presents an approach functioning as an alternative to that prevailing dichotomy.

**Keywords:** technology; intentional community; ecovillage; commune; modernism

## 1. Introduction

Ecologically minded activists have long sought to both conceptualize and enact visions of a "sustainable" world and "eco-friendly" lifestyles, often leading to contradicting claims about what those might look like or how we might get there. Will technology save us, or is the pace of innovation leading to the ultimate demise of civilization as we know it? Is individual action or a grand policy initiative the most effective path? Thinkers, scientists, and activists can seldom agree on answers to these cosmic questions, which adds up to a pluralistic and often contradicting ecological movement.

Experimental answers are found to differing perspectives on these questions in a variety of formats from simple living movement adherents who eschew technology as much as they can, to large corporations and non-profits who offer high technology as a way of engineering the unsustainability out of current consumption patterns. This article takes a microsociological approach to the question of sustainability, lifestyle, and technology use by examining a single "ecovillage" that is part of a worldwide movement. The ecovillagers maintain a direct relationship to the concept of ecological sustainability in their consideration over what technologies to use. They want to achieve an ecologically sustainable lifestyle as their central mission and have created a lifestyle around this mission that treats technology in a distinctively non-modern fashion.

An "ecovillage" can conjure imaginations of an idyllic place, one devoid of the interferences of modern technology, where a person can go to retreat from and reflect on nature. Yet it is not so easy to escape the rest of the world, especially when the overarching goal is to have an influence on it and change its orientation to be more sustainable and ecologically conscious. The purpose of this paper

is to examine the relationship between technology and lifestyle through the narratives of progress using a qualitative case study conducted at an ecovillage. Ecovillages are a source of technological innovation, but where the ecological motive replaces modernist ideas of growth and innovation as the primary driver. This is observable in how they collectively submit to communal covenants and a collective lifestyle that accommodates diverse attitudes towards the use of technology while reorienting its purpose. The case presented here will be used as a means of furthering conversations about that ecological motive and developments in theory over human–technology relationships in sustainability-minded communities. Such relationships are complicated, and require a theory based on rich, nuanced data about the lived experiences of people who are attempting to live an ecologically conscious lifestyle.

## 2. Literature Review

Intentional Communities [1] are a form of human gathering that is characterized by communal living, with "a minimum of three adults who share a common dwelling, household duties, meals, belief system, provide emotional support for one another, and identify themselves as a communal household or commune." [2] These are often hard to draw clear boundaries around since they have taken on so many different forms, as Miller [3,4] has pointed out, but nonetheless tend to have certain shared aspects useful for the purpose of analysis: a shared purpose or vision, close physical proximity, a financial or material sharing arrangement, and at least five adults not all related by marriage living together. ICs have existed in different forms and for different purposes going at least as far back as the Protestant Reformation and have served as small laboratories of social change. Early Puritan Colonies in North America and the first Mormon groups have been characterized by scholars as ICs [5,6]. Likewise, some more famous communal groups like the Shakers, Amanans, Hutterites, Fourierists, and Moravians have dotted the landscape of mostly the United States but also in other places colonized by European peoples. The IC is perhaps most iconically known in much of the Western imagination from the countercultural communes that sprung up in the 1960s and 70s, though they will also be recognized in the Kibbutz movement in Israel or various other religious and secular social experiments—anywhere small groups of humans settle together on purpose in order to achieve their collective vision for society through the mechanism of communal living. Ecovillages in particular, though, seek to incorporate sustainability as a central feature of the group's lifestyle.

In 1991, Robert and Diane Gilman, in a report to the Gaia trust, described a new form of IC. These "ecovillages" that emerged around that time are communal experiments in "human scale full-featured settlement in which human activities are harmlessly integrated into the natural world in a way that is supportive of healthy human development can be successfully continued into the indefinite future." [7] These settlements have become an increasingly popular form of IC since the 1990s, in part because of the increasing mainstream prominence of the global environmentalism movement [8]. Newer villages such as the case study presented here have come about, but older communes who have persisted since the 1960s and 1970s such as Twin Oaks and The Farm in the United States or Findhorn in Scotland have adopted the title of ecovillage as the global environmentalism movement both became intertwined with other countercultural forms and then eventually became more mainstream. Ecovillages can form in both urban and rural environments, but all have some degree of separation from the larger society regardless of whether that is social or geographical. This "estrangement," as communal studies scholar Lucy Sargisson puts it, is a means of providing distance from the rest of society and reinforcing membership to a smaller community that generally shares a set of ideals [9].

This paper shows in a case study how ecovillagers work to eliminate the notion of linear progress in their efforts at community building and eschew the contrast between modern and primitive technologies in their practice of building the community. Existing within a larger society means they must acknowledge and use the prevailing language of modernism while resisting it in order to give their own accounts of time times. Ecovillage life presents an intentionally alternative conception of progress and the meaning of community [10], and "by creating a certain way of experiencing the world,

in addition to promoting an intellectual understanding of the reasons for living this way, ecovillages represent the synthesis of knowledge and action, theory and practice." [11] (p.20) This is historically similar to other forms of ICs, for whom Hall [12] says that " . . . the revolution, whatever its nature, is carried out in everyday life . . . Alternative communal groups thus give new accounts of the times – both of the old world of the established order and of the new communal world" (p. 43)

In other words, by giving "new accounts of the times," ICs create new definitions of the past, present, and future that help to re-center their member's narrative realities around the core mission of the community. To a large extent, this is because ICs tend to survive as a balancing act—they exist by virtue of their mission but must contend with the rules and norms of the larger society in which they reside [13]. By rejecting a default notion of progress offered by the larger society, IC members seek estrangement from it and instead attempt to live the embodiment of their own [9]. It also means that these efforts are in relationship to the excesses of contemporary society and in resistance to them, thus positioning communitarians (and ecovillagers) in an oppositional relationship to them.

Rejecting prevailing accounts of the times is a project that extends beyond the work of ICs and is backed by a philosophical tradition seeking to undermine the epistemology of modernism itself. Philosopher Bruno Latour [14] leads the way in this endeavor with his argument in "We Have Never Been Modern" that the work of modernism and—as is argued here—the work of mainstream notions of progress has been an exercise in purification between ancient and modern, or as a synecdoche for an operational dichotomy between nature and society. Latour says, "I may use an electric drill, but I also use a hammer. The former is thirty-five years old, the latter hundreds [or] thousands. Will you see me as a DIY expert "of contrasts" because I mix up gestures from different times?"[14] (p. 75) For the ecovillage form that we are focused on here, countering that project which has sought to separate the modern-ness of humans from the primitive-ness of nature is especially important, as ecovillagers have been shown to "make explicit the connectedness of the individual to the social and ecological worlds" (p. 332) in the pursuit of their version of intentional community [1].

Chitewere [15] describes the ecovillages movement as a rather privileged endeavor due to the steps that people must take to remove themselves from an unsustainable society. According to her research, ICs are consistently comprised of individuals with middle to upper middle-class, predominantly white, and well-educated backgrounds. The data from the case study in this paper does not dispute this, though it would be inaccurate to describe it as completely homogenous in any of those aspects. Ecovillagers are, in Miller's estimation "as passionate about their environmental convictions as medieval monks and nuns were about their religious beliefs." (p. 145) [4] Therefore, rather than ecovillage life being something well suited to middle class white people in particular, it is their background that often leads them to seek out ecovillage life because it is possible, often with little to no risk if they are not able to make it work.

Perhaps because of this relative homogeneity and privilege, or perhaps despite it, ecovillagers tend to be quite happy with their lives. According to a survey by Grinde, et. al. [16] using the Satisfaction with Life Scale [17], ecovillagers are happier than almost any other measured demographic in the United States and Canada. This can be explained, at least in part, by features commonly found in groups living communally, like consensus decision-making processes that give members a sense of buy-in to the community's governance processes [18]. An expanded sense of community ethic and satisfaction with their ability to participate in community decision making [12] may also contribute to this.

Previous research has shown ecovillagers' approach to be successful in accomplishing the mission of sustainability. Many ecovillages have embraced alternative designs and successfully reduced their ecological footprint by a significant margin compared to peer citizens [19–21]. Recent studies in ecovillage-style communities like Cloughjordan in Ireland [20], Findhorn in Scotland [22], Sieben Linden in Germany [23], and Piracaia and Porongaba in Brazil [24], among others, show that this form is relatively widespread and successful in reducing the ecological footprints of members when compared to their compatriots. They also show that ecovillages have developed "narratives of change" in service

to countering the neoliberal narrative that positive change must come through social innovation [25]. The environmental success of the case study contained in this paper is below, in the Methods section.

That they serve as centers for alternative narratives of progress also means that ecovillages can serve an educational mission, as a node from which ideas are developed and dispersed through ecovillage educational endeavors. These "transformative sustainability education" programs [22] have served as an avenue through which non-ecovillagers may learn more about sustainable behaviors embodied in their alternative account of the times. Alternative accounts, too, require what Papenfuss and Merritt describe as alternative pedagogies—ritual pedagogies, pedagogies of story, and collaborative pedagogies—such as those found in an ecovillage education program at Findhorn [22].

Despite their alternative conceptions of time, progress, and community, ecovillagers must create their experiment in the confines of the social and political world which they either seek to be separate from or to influence and change. Additionally, they submit to their own collective restrictions put in place as a means of trying to accomplish their sustainability mission [3,4]. Thus, ecovillagers' individual relationships to technology are dictated by the confluence of several social forces: what they have access to via the socioeconomic standing of their members, their own conception of progress against a collectively defined mission and what technologies are appropriate within that vision; and what they are allowed to use under state and local regulations. As the data from the case study presented below shows, these tensions are present throughout their deliberation and application of various technologies deployed to achieve a more sustainable lifestyle. Yet the varied standpoints in relation to sustainability all add up to a singular ethic of ecological innovation which stands in contrast to modern society that prizes economic innovation.

## 3. Methods

### 3.1. Site

Data for this paper comes from an ethnography carried out at an ecovillage named Dancing Rabbit (DR). DR is situated in rural northeastern Missouri, shown in Figure 1 below. It is a place where their lifestyle and eco-consciousness stands in stark contrast to countervailing local currents of evangelical Christianity, modesty standards characteristic of the large Mennonite population in the area, and overall cultural conservatism of the region. Where "Rabbits" (as they call themselves) may swim naked in their pond or openly discuss their bodily functions as they compost their feces, when going to town for groceries or to eat out they alter their dress and speech to be more tolerable to the local Mennonite-run businesses. The nearest town has a population of 110 as of the 2010 census, and the county seat has only 1800, which underscores just how "out there" the ecovillage is. Yet the founders of the community picked this site for just such reasons of location; in rural Missouri, there is a lack of building codes and other governmental interferences, which has allowed them to deploy many alternative technologies described below, where a more urban or suburban location might not have sufficed.

### 3.2. Data Gathering

As part of an ongoing research project, I lived at the site for eight months observing and participating in the activities of the ecovillage. Gathering data during this time involved participant observation in everyday activities of the approximately forty-five adult Rabbits, including such activities as constructing homes, collective meal preparation and consumption; interacting with visitors and guests to explain the Rabbit lifestyle; and sharing in leisure activities like drinking beer and talking about current events at the on-farm bar/bed-and-breakfast.

Data collection also consisted of a series of eighteen semi-structured ethnographic interviews on the Rabbit lifestyle that ranged in length from fifteen minutes to an hour in length. Questions for these interviews fell under a few key themes relevant to this project: social norms, Rabbits' individual views on the community's mission, and their thoughts on human–environment relations. Due to the nature

of ethnographic interviews, these questions were not the same from person to person since the author's knowledge of respondents' different backgrounds allowed for more tailored inquiry. However, some examples of these questions included "how would you describe your relationship with nature?", "I'm interested in learning more about what things are not accepted as everyday practice in the community, but might be in the rest of society. Could you tell me more about those?" and "what would you say your goal is in living intentionally here? Will that goal ever be accomplished?"

**Figure 1.** Map showing the location of Dancing Rabbit.

Finally, further qualitative data for this paper was gleaned from members' blogs and other public writings. In particular, data from their official blog is referred to as the "MemDem" throughout this paper, an internal nickname drawn from early in the community's history where they regularly published a column in a local newspaper named the Memphis Democrat which is now primarily produced in digital blog format just for the community's followers. Each of these datasets was compiled and coded separately. I coded the interviews in the qualitative analysis software NVIVO, and I searched the MemDem collection using keywords related to technology. Then, I read through each collection several times, inductively generating themes related to technology at the ecovillage. Throughout this paper, quotes from these interviews and published writings are used to compliment and explain key themes observed in the course of ethnographic data gathering. Many of the names have been changed to protect participants' privacy.

*3.3. A Note on Ethnographic Methods*

A large part of DR's mission, in addition to sustainability, is outreach. As an example, they have a community non-profit named the "Center for Sustainability and Cooperative Culture at Dancing Rabbit" (CSCC) which is used to recruit visitors and potential new Rabbits as well as produce media supporting the dissemination of their lifestyle to the world. This stands in contrast to many other ICs who opt to completely retreat from society, rather than seek estrangement, in order to better live their ideals. DR's outreach has thus led them to be scrutinized and publicized by a variety of outsiders from television producers (Morgan Spurlock filmed an episode of his show "30 Days" at

DR), documentarians, student groups, environmental engineers, academics, and a host of everyday visitors who come to see what ecovillage life is all about. Beck and Ormsby [26] found DR to be effective in accomplishing several parts of their mission, namely in establishing effective and functional self-governance as well as a gender and spiritually inclusive culture.

Most germane to their goal of living sustainably, Jones [19] and Lockyer [21] conducted an audit in cooperation with the CSCC of the consumption patterns of Rabbits and found promising results to the group's endeavor: Rabbits consumed, on average, 6% the total vehicle fuel, 14% the electricity, 19% the fresh water, and 5% the natural gas as the typical American and produced 47% as much municipal waste. This evidence strongly supports a hypothesis that their lifestyle is far more sustainable than the typical person living in the U.S. As each of those studies and my data suggest, Rabbits have reduced their ecological footprint in a large part thanks to their thoughtful engagement with different types of technologies. Lockyer borrows the phrase "degrowth" to describe this phenomenon—a utopian discourse of post-carbon, post-growth transition that emphasizes forms of development under an emerging cultural logic of sustainability over economic growth [21] (p. 521).

There are many uses for, and limitations of, the sort of ethnographic research performed for this study. It is an inherently reflexive exercise, one that Burawoy calls a "Siamese twin" to positive science research [27] due to their complementary strengths and weaknesses. This means that ethnographic studies, by nature of their highly focused and specific object of study, often cannot be reproduced across groups or time. But that is not their purpose. Rather, the goal is to provide context for future positive science [28] which often lacks the reflexivity ethnography provides.

In arguing for the importance of ethnographic research, Jeromack and Khan [29] have noted that people do not always do what they say and that self-reported behaviors found in positivist studies often fall prey to the attitudinal fallacy which assumes that peoples' attitudes and actions are the same. So, while positivist measures are often useful because they are inherently more reliable and replicable, it is only when paired with the reflexivity of ethnographic methods that we can be certainty of such reliability and replicability in generating rigorous theoretical understanding of social phenomena. In thinking about the effectiveness of this and other ethnographic studies, then, it is most useful to see the methods as a means of "extending out" to theory rather than testing and refining it [27]. By gathering accounts of both what people say *and* what they do, researchers engaged in ethnography are more situated to generate theory that is firmly grounded in the social reality of their subjects [30].

The data below, then, is in many ways an elaboration of both Lockyer and Jones' works. Where they showed that DR consumes far fewer resources than the typical US citizen, I show the narratives and practices developed in a tightly cohesive and intentional group that make those reductions possible through a prevailing ecological motivator. While there are many different success stories of ecovillages attaining vastly more sustainable lifestyles, there are many different ways they go about it. The results I consider to be a step in establishing a link between lifestyle and sustainability, albeit one link of many possible sustainable lifestyles. The data informs the replicability of ecovillage experimentation with alternative technologies, and moreover with alternative conceptions of technology and progress.

## 4. Data and Results

In this section, the relationships with technology and the ecological motivator are discussed across several categories where technology and the sustainability motive come in to contact, including dwellings, resource consumption, digital presence, and examples of how these add up to pluralism in the use of technology. In all of these, Rabbits are restrained through their collectively agreed-upon six ecological covenants and eight sustainability guidelines (shown in Figures 2 and 3) that prioritize sustainable practices but leave some room for interpretation in practice. Some examples of these include a ban from personal ownership of vehicles, a restriction on the use of fossil fuels for any activity other than cooking, and a requirement that all construction materials be sustainably sourced. These are also discussed further in the section on resource consumption below. Ma'ikwe writes in the MemDem of this diversity in attitudes, stating that "[w]e have people who work on computers, and people who

work the land. Pick pretty much any sustainability issue, and we are probably mindfully embodying different approaches." [31]

1. Dancing Rabbit members will not use personal motorized vehicles, or store them on Dancing Rabbit property.
2. At Dancing Rabbit, fossil fuels will not be applied to the following uses: powering vehicles, space-heating and -cooling, refrigeration, and heating domestic water.
3. All gardening, landscaping, horticulture, silviculture and agriculture conducted on Dancing Rabbit property must conform to the standards as set by OCIA for organic procedures and processing. In addition, no petrochemical biocides may be used or stored on DR property for household or other purposes.
4. All electricity produced at Dancing Rabbit shall be from sustainable sources. Any electricity imported from off-site shall be balanced by Dancing Rabbit exporting enough on site, sustainably generated electricity, to offset the imported electricity.
5. Lumber used for construction at Dancing Rabbit shall be either reused/reclaimed, locally harvested, or certified as sustainably harvested.
6. Waste disposal systems at Dancing Rabbit shall reclaim organic and recyclable materials.

**Figure 2.** Ecological covenants of Dancing Rabbit.

1. Dancing Rabbit will look holistically at the issues of sustainability to create a sustainable culture that takes into account all impacts of its actions and acts to preserve the Earth for the future.
2. Dancing Rabbit will strive to rely only upon renewable resources, and to use them at a rate less than their replacement.
3. Dancing Rabbit will try to understand and minimize its negative impact on global ecological systems.
4. Dancing Rabbit will attempt to preserve and rebuild healthy ecosystems and have a positive impact on biodiversity.
5. Dancing Rabbit will try to create a closed resource loop where byproducts are reintegrated as useful resources, thus attempting to minimize waste products, especially those toxic or radioactive.
6. Dancing Rabbit will try to avoid exploiting people and other cultures.
7. Dancing Rabbit will strive to achieve negative population growth from reproduction.

**Figure 3.** Sustainability guidelines of Dancing Rabbit.

Throughout these sections, data is presented to show how Rabbits' attitudes differ towards the use of technology in achieving the degrowth mindset while simultaneously adding up to an alternative "account of the times" that rejects modernist assumptions about technology in favor of an ecologically minded one. In other words, Rabbits have different mindsets on the use of technology and draw from many different generations and traditions to achieve ecological sustainability, but all agree that sustainability is the most important outcome. These are divided into three major themes across which attitudes about technology are particularly salient: how they construct their dwellings, how they manage resource use, and how they differ in attitudes towards technology in general.

*4.1. Dwellings*

One feature of DR's location as a retreat to a rural area is that they are not beholden to any sort of building codes. This means that they are free to construct alternative forms of housing and experiment

with materials or forms that would otherwise be unapproved in an urban or suburban settlement. In an interview, Rabbit Hassan explained this motivation to me, stating that " … we're able to do natural building without codes because there's not code limit in rural Missouri. A lot of communities have code restrictions and so a lot of what they do as a community goes under the radar of public eye because they don't want to get busted for this, that, or the other thing. "[32] The dwellings they build are often at the intersection of ancient and modern technologies, though some are entirely primitive and some entirely modern. Their ecological covenant that "lumber used for construction at Dancing Rabbit shall either used/reclaimed, locally harvested, or sustainably harvested" applies here, as do many of the sustainability guidelines surrounding resource use. Lumber, though, is not the only material that goes in to building a dwelling, so the rest of the construction process offers room for interpretation through each member's own ethic of technology use.

Schelly [33] describes how "[t]he material arrangements and social practices at DR are organized based on a dance between two set of values: one that recognizes the symbiotic relationship between humans and the natural world as well as humans with one another, on the one hand, and one that values independence, on the other." (p. 96) Her data, gathered contemporaneously to that used in this article, is focused on the built landscape of DR, and explains how their dwellings constitute a form of resistance to mainstream growth narratives, while here the focus is on how technology is embedded in those dwellings. The value system behind the construction of dwellings at DR reflects an interpretation of the times consistent with the literature above, one where the modernist project of purification separating primitive nature from advanced humans is on the block under the Rabbits' splitting maul.

Perhaps the most salient way that ideas about technology use are filtered through this lens of resistance can be seen in the many dwellings that Rabbits have built out of straw bales. Nineteen of the thirty-two dwellings in use at the time of data collection were built in this style, which is an ancient technique that many modern builders concerned with sustainable living have adapted to contain several modern technological innovations. The typical pattern for building one starts with the laying of rigid polystyrene insulation boards on top of a levelled gravel surface and the pouring of a concrete pad over top of those. Large timber beams hewn locally are erected and sunk in the ground to outline and support the frame of the house, which is then filled in with straw bales purchased from neighboring farms who have assembled them with modern baler equipment. Holes are cut in the bale stacks for doors, modern high-efficiency windows, and utility connections. Then, in the most laborious step, builders apply a mixture of mud, straw or cattail fluff, and sand termed "cob" to both the inside and outside of the house to finish the walls. Straw bale buildings will often have many layers of cob applied to them, as this material allows for curved edges other interesting wall features. The components of cob act similarly to the components of concrete: the mud binds like cement, the sand is the same bulking agent in both, and the fibers from straw or cattail act as rebar that prevent the coating from cracking as it dries and shrinks in mass. On the outside, more sand is used for a rougher feel and often a lye paint or linseed oil is applied for both a neat finished look and weather proofing. On the inside, less sand is used in the mix and the straw is replaced with the finer cattail fibers for a smoother and more finished look. Since cattail requires collection by hand and is only seasonally available, it is often only used for the final layer out of many. At the conclusion of our interview, Hassan departed saying "I have to collect cattail fluff because it's that time of the year right now." [32]

Other straw bale structures are built with various combinations of ancient or natural techniques combined with modern tools or aesthetics. Kyle built an earth-contact straw bale home using high-end Milwaukee brand tools, which he was very excited to show off and describe during his interview. On the other end of the tool spectrum was Tadla, whose house was many years in progress (with many still to go) under manual labor using traditional Japanese woodworking hand tools and joinery. Other carpenters and homebuilders like Hassan and Sambucus leaned heavily on old-fashioned woodworking tools like the hammer, mallet, chisel, and manual planer, though they also owned powered drills and saws I saw them use on occasion. Many of the straw bale houses contain a "truth

window," not to combat any notion that there might be lies contained in the walls but to show guests the materials from which the house is constructed. Usually taking about one square foot of wall space, these is a small glass panel embedded in the normally smoothed cob walls that is used to reveal the straw bales contained underneath.

As the village has aged so have those houses, and some of their problems have become more apparent. Missouri is quite humid, and that humidity penetrates cracks in the cob to make the straw bales expand and contract as the temperature changes, thus worsening the cracks and forcing repair. It can also lead the straw bales to rot, which is a far worse problem because then they must be torn down and replaced completely if they lose their insulating and structural capacities. Therefore, many Rabbits in straw bale homes have taken to using air conditioners, previously thought an extreme luxury for a straw bale house that passively keeps the temperature below 25 °C even on 35 °C or hotter days, because they will help extract moisture from the air and reduce the chances of it rotting their homes. Alyssa bemoaned this reality to me one day while we were preparing dinner in her hot kitchen, that she had lived at DR for a decade with no air conditioner but felt she had to get one after discovering some rot through her truth window. [34]

Lop wrote in a post in the MemDem explaining how he intends to resist adding an air conditioner to his house because "[a]ir conditioning comes in many forms. My current preferred format is called swimming. It isn't a perfect system, but all the energy it takes to create the pond has already been spent."[35] Kyle likewise expressed in his interview that air conditioners should be accepted but that they were not ideal: " . . . and you see more air conditioners, and more power tools. These are things that I use myself. It would be nice to see DR hold on to a culture of radical simplicity and simple living. I think the biggest challenge to that is our explicit desire to grow, and I think it's harder to grow while keeping a radical culture of simplicity because it's not what we as creatures want. We might say we want it, but often it's not what we want." [36] To Kyle, for DR to succeed in their mission of sustainability means embracing many approaches, even ones he might not prefer himself.

Most of these houses contain modern electrical wiring for lights and outlets, and most of these are hooked up to a solar panel array on the roof or to the internal micro-grid discussed below. Homes that do not have solar panels often had "living" roofs which were constructed with a base layer of EPDM lining (an artificial, petroleum based, heavy-duty rubber material) then topped with soil so that the resident could have a garden above them. In an interview, Sharon summed up this approach of using as many natural materials in her house as possible while recognizing the benefits that some modern materials like an EPDM bring as an expenditure of "junky credits." These credits, she explained, were her lingo for describing an appropriate if regrettable use of manufactured materials because it would facilitate other, more sustainable behaviors like roof gardening and would help her house built with mostly natural materials last longer. [37]

Other houses were built with modern conventional materials and designed to last as long as possible to offset the perceived ecological impact of those. In order for Tereza to have her house made of today's popular materials like dimensional lumber and concrete board siding, it was necessarily small and heavily insulated. Even though it took more resources to produce, she was comfortable with that because it would outlast its peers and use less energy by virtue of needing to heat or cool less space. It was also built on concrete piers a couple of feet above ground such that it could be easily picked up and moved to another location should she eventually sell it, which she expected would increase its reusability since it was not locationally bound as most houses are.

In addition to the question of materials is also one of design. Buildings at DR are typically highly planned endeavors, making use of passive energy saving techniques and south-facing roofs to capture sunlight. Several of the structures at DR were architecturally interesting in one way or another, containing features like awning windows near the ceiling that let hot air rise out of the building and cool air to rise from the ground-contacting concrete pad, or windows that only allowed sun in when it was lower in the sky during the cold months and not during the warm months. Others are feats of straw bale engineering including the two-story, six-bedroom Skyhouse apartment complex, which the

founders built complete with its own solar array, battery system, and water capture cistern and filter system, making it a nearly self-sustaining dwelling.

The completion of Skyhouse required many hands, and an expensive architectural contract. But, as with many of the dwellings at DR, it was an experiment as much as it was a statement about their ecological values. Sable, who also lives in a straw bale structure and has worked as an architect on many projects at DR including Skyhouse, described her experience building in this way: "I would say sustainable architecture and working together as a team of people that don't necessarily have the answer in advance - we're learning as we go along, and we have passion and excitement and camaraderie as our driving form."[38] While Rabbits could each be found to have different philosophies on building, as most individual homebuyers or builders do, there was nonetheless an overarching philosophy that the use of modern building materials should be integrated with natural materials, and each could be deployed in useful, sustainable ways.

*4.2. Resource Consumption*

An overarching goal at DR in line with their mission of sustainable living is to do more with less, or to live life using fewer resources than the typical American. Most of the sustainability guidelines, like "Dancing Rabbit will strive to rely only upon renewable resources, and to use them at a rate less than their replacement" and "Dancing Rabbit will try to create a closed resource loop where byproducts are reintegrated as useful resources, thus attempting to minimize waste products, especially those toxic or radioactive," for example, are directly supportive of resource use reduction. This section discusses some of these aspects to the lifestyle at DR which have contributed to the overall sustainability ethic. Rabbits fully understand that this is part of what they contribute to by joining DR. Danielle put her thoughts on this succinctly in an interview, stating that "[e]covillages and intentional communities are … new systems, testing prototypes for human cultures that are thriving, just, and sustainable. Dancing Rabbit Ecovillage residents place environmental impact at the core of our considerations … " As shown above in both Lockyer [21] and Jones [19], placing the environment at the "core" of their considerations for resource use has panned out very successfully, as DR like many other ecovillages has managed to reduce their consumption dramatically when compared to the typical U.S. citizen. These are thanks to a combination of the ecological covenants and sustainability guidelines, lifestyle changes, and the deployment of alternative ideas about technology.

One practice relating technology to consumption is the use of permaculture by many of the Rabbits. Permaculture is a somewhat recent practice that can be traced back to the 1970s but is based on far more ancient methods and ideas for designing sustainable local food production regimes. Yet it was also developed with a mind towards emergent ideas of human design in the natural landscape. David Holmgren, one of the early pioneers in the permaculture movement, defines it as "Consciously designed landscapes which mimic the patterns and relationships found in nature, while yielding an abundance of food, fibre and energy for provision of local needs." [39] DR holds an annual workshop on Permaculture, bringing in experts and paying students to both learn from the homestead permaculture gardens surrounding many Rabbit dwellings and help construct them further. For Rabbits, in this sense, degrowth is a process of growing while deconstructing prevailing definitions of what it means to grow. Instead of agriculture through machinery-intensive methods, Rabbits practice agriculture through design-intensive ones. As Lilac reflects in the MemDem, "[at DR] … permaculture is our relationship to water, sun, buildings, food, health and ourselves, in convergence with common sense, indigenous wisdom, and appropriate technology for greater food yields, for natural systems that are less work to maintain and that restore local environments."[40]

This notion of "appropriate technology" is something under debate and individually determined at DR, though they have collectively agreed to several specific goals and limitations through their covenants and guidelines. One covenant at DR states that "[a]ll electricity produced at Dancing Rabbit shall be from sustainable sources. Any electricity imported from off-site shall be balanced by Dancing Rabbit exporting enough on site, sustainably generated electricity, to offset the imported electricity."

To that end, in 2011 the ecovillage built an internal electric micro-grid named Better Electricity for Dancing Rabbit (BEDR) to facilitate the home use of solar panels and eliminate the extra expense of buying batteries for a comprehensive off-grid solar system. Many of the houses have solar panels made possible by this system, with which they also produce a surplus of electricity during the day and draw from their hookup to the larger grid for power at night. At the time of data collection for this article, Rabbits exported approximately 90,000 kWhr of electricity during their surplus times and imported approximately 51,000 kWhr [41], a notable net export of their sustainably produced electricity to a statewide system mostly reliant on coal.

Not everyone is hooked up to the grid. Some Rabbits, like Widder, had purchased batteries for a household solar system before the advent of the BEDR grid and plan to see them through to the end of their decades-long life cycle before hooking up to the rest of the community. Others like Sparrow or Lop have actively avoided hooking their homes up to BEDR because they prefer to have minimal electronics in their lives but nonetheless supported the presence of BEDR because it dramatically lowered the cost barriers for renewable energy in the village through collectivization. In both cases, their decisions seem to be informed by the sustainability guideline that "Dancing Rabbit will try to create a closed resource loop where byproducts are reintegrated as useful resources, thus attempting to minimize waste products, especially those toxic or radioactive."

Another covenant at DR states that "[w]aste disposal systems at Dancing Rabbit shall reclaim organic and recyclable materials," which has translated into a complete absence of flush toilets. Rabbits and visitors are expected to urinate outdoors among the grasses and trees, and to defecate in designated composting containers. To many of the visitors that come and are used to being able to flush their waste away, this is a distinctly anti-modern approach to waste management. Yet, it is an integral part of the lifestyle at DR and their notion of ecological progress. Conversations between Rabbits and visitors about the composting of human waste (termed "humanure") were abundant: how it worked, where the composted waste would be applied once fully "rested", and how much water they saved by never flushing.

This composting takes the form of a "pooping in a bucket" approach to waste management, a phase that was common parlance in my observation. Rabbits using this approach employ a standard 5 gallon plastic bucket installed in a wooden box with a toilet lid for their bowel movements and cover their leavings with sawdust to desiccate the leavings so as to inhibit bacteria growth and minimize the smell. When the buckets become full, they are dumped into large compost piles several hundred yards outside the village to further preclude the spread of feces-borne pathogens. Rabbits then let the piles "rest" for several years and have been using the resulting compost to fertilize trees as part of a reforestation program they have been conducting on the land. The buckets are "modern," so to speak as they are made of plastic, but the whole premise of the endeavor is a rejection of a larger social consensus that modern plumbing is a form of progress. Rather, Rabbits view flush toilets as a wasteful abandonment of useful nutrients that could be saved to fertilize their land and the ecological motivator incentivizes the capture and reuse of them.

This mix of new and high-tech with old and low-tech goods could be found among many of the deliberations and decisions at DR. Loren spoke of this phenomenon at two different points during her interview. At one point, she said " . . . my main attraction for visiting Dancing Rabbit over other places was because I like technology. I think that information is a huge thing, and our access to it through the internet. In being able to help other people to live better on earth it's really important for us to find and apply new technologies that make it easier," [41] belying DR's openness to newer, digital technologies. Likewise, she said that "[t]his could be a place where you come to see all kinds of inventions, you know? Pedal power to run your washing machine, things for people of every income that could be applied," [42] explaining her standpoint where newer or more advanced did not always meant better. While individual Rabbits hold different opinions and proclivities about the use of technology in lowering their ecological impact, the Rabbit lifestyle is nonetheless one that has adopted *both* ancient and modern technologies without discrimination, thus creating their own notion

of progress counter to the prevailing one. The more important distinction for them is whether using any particular technology will help to decrease their ecological footprint and reduce resource use.

### 4.3. Diverse Attitudes Towards Technology

From its very formation, contemporary technologies were key to the community's survival. DR was founded by web developers, who came to Missouri from San Francisco and eventually chose the site where the village is today based on the fact that it had a decent internet connection for them to run their business "Skyhouse Consulting" in addition to building an ecovillage. The name is no coincidence either, as the Skyhouse consultants were also the ones who built the Skyhouse straw bale apartment building. Nik writes in the MemDem that "using technology and making the community a more prosperous place through engineering and science also has strong roots here, back to the founders. They were not cart-and-donkey back to the landers, but students and computer programmers who cared deeply about the future of the planet. Now there are Rabbits with a cart and a donkey, but that cart was still beautifully engineered!" [43] In fact, the availability of internet early in the community's history helped it blossom as a source of revenue and recruitment, even as those that came to the ecovillage often desired an approach to ecological sustainability that eschewed such digital technologies.

There is a spectrum between love and hate of the digital technologies when it comes to the internet at DR, though even those who hate it still find need of it on at least a weekly basis. At DR, as it is for most of the rest of the society in which they live, barriers to survival are higher for those not engaging in digital technologies of some kind. Rabbits rely on email for communication with friends and family or searching for work. Yet, like the rest of society, there is a gradation between the acceptance of technology from those who would never use it if they could to those who would use it for almost everything if they could. These tendencies are not monolithic, but what is monolithic is that this spectrum of tastes rests within a commitment to sustainable practice.

On one side of this spectrum, Rabbits like Havana and Lop preferred to get their news and entertainment from the radio or printed newspaper. In rural northeast Missouri, where the population is much older than the U.S. average, this is still fairly easy for them. This cohort found newer technologies to be an unfortunate distraction, like Dennis who said in his interview that "[w]hether it's sports or addiction to video games or to diversions, these entertainments are meaningless and excessive in order to divert people from real meaning in life." [44] Alline related somewhat similarly to newer technologies, saying in her interview that "I'm not boycotting iPhones, I just don't have one yet. I don't like what it's doing to the wide culture. I hate when you go somewhere and no one's looking at each other, they're all checking their phones. I just think that's appalling. I'm 58, so I probably am just a big crank." [45] That she mentions her age in discussing her relationship with technology is telling, as I observed this spectrum of embracing to rejecting new technologies to have a moderate connection to the relative age of individual Rabbits.

For others in this camp, becoming a Rabbit was an opportunity to engage less with such technologies. Sparrow found living at DR a refreshing opportunity to unplug from a previously more screen-dominated life. She said in her interview that she keeps up with the news far less since moving to DR, and that

> "[n]ot watching the news is a byproduct of not watching TV. Something that hugely changed for me - it was really easy to get inundated with the message that the state of the world is shit. That is what I was being told constantly and being surrounded with, even by the people around me. Moving here, all the sudden I was meeting people who were coming through from other communities. Just meeting and getting to know all of these networks, this communal network that all of the sudden I was like "oh my gosh, duh!" There are millions of really awesome people in the world, and what I was just seeing when I was living in the city with my partner and watching TV was my bubble, which was filled with a lot of pessimism." [46]

Leaving those amenities behind was part of the lifestyle for her, though she would still use a computer several days a week for correspondence and other business.

On the other side of the technology spectrum, digital enthusiast Sallander spends most of his day on the computer running an SEO (search engine optimization) consulting firm. Cob is similarly on the computer a lot doing non-profit consulting as well as doing the bookkeeping for several village-owned entities like their land trust. Even Loren, who was noted above for her enthusiasm of pedal-powered appliances, runs a small computer-repair business at DR. Yet most Rabbits fall somewhere in the middle of these extremes. They lauded the digital technologies that helped them accomplish their mission while expressing distaste for any excesses that distracted from it or made it more difficult.

Ma'ikwe described how living at DR has allowed her to put some of her lifestyle concerns like minimizing consumption and reducing fossil fuel usage in the background because they are accomplished as a matter of course by living in the ecovillage. But, being a self-identified activist, she sees using the internet as a launchpad for talking about that lifestyle with others:

> *"**Ma'ikwe**: Living here has also freed me up to have more national and somewhat international connections. I'm partnering with and encouraging good things to be happening at a much bigger playing field, where I think my activism before I moved here and got my own scene set up was very locally oriented. Generally, I wasn't doing any national networking stuff.*
>
> ***Interviewer**: Interesting. Moving to rural Missouri has freed you up to do more national and international collaboration.*
>
> ***Ma'ikwe**: I think part of that is just the development of the Internet too. Some of that might have happened anyway."* [47]

Ma'ikwe is not ambivalent, but rather embraces all of the above approaches to technology that approximate a median to how Rabbits think about it. While living in the straw bale house she constructed, she spends much of the day on her laptop collaborating with other communitarians across the world through video chat software and email. Yet, she also expressed appreciation for living in a village where she did not have to struggle with reconciling working digitally and living more sustainably.

There are some technologies that one can find nearly ubiquitously throughout the ecovillage. Refrigeration, common to much of the rest of society, is similarly found in every kitchen at DR. Sallander and Sparrow, while taking different tacks on the importance of laptops to everyday life, shared a passion for the use of portable electricity usage monitors—the kind one plugs in to an outlet before plugging their appliance in to it in order to determine how much electricity it uses, measuring its efficiency. Most people grow their own food at least a little, and therefore need to use a shovel.

In this spectrum and in all of the above approaches, we can detect how DR embodies an alternative notion of progress predicated on the ecological motive. This alternative conception of the world, where the use of digital technologies at DR has followed their development in the larger world, rejects the notion that newer technologies are inherently better or more suited to their mission. Instead, the appropriateness of any technology is tested against whether it helps Rabbits to lead a more sustainable lifestyle. Rabbits are not indifferent to technology but have carefully considered and diverse relationships to it. These varied technologies help different Rabbits achieve their vision of a sustainable world while eschewing a modernist paradigm that would require them to swear allegiance to the latest technology as the solution to minimizing human impacts on the environment or swear off it as anathema to a sustainable lifestyle.

## 5. Discussion

DR has been built in a social world where the advancements of humans are typically treated as an improvement over nature that separates humans from it. As the data above shows, the countervailing current at DR is to prize advancements that put Rabbits in closer contact with their narrative of nature, prioritizing an ecological motive that places both ancient and modern technologies as potential partners

in development rather than as antagonists. While individual Rabbits have varying approaches and relationships to technology, they have a strongly shared collective notion about its role as furthering their vision. By joining the ecovillage, they have entered a collective effort bounded by shared covenants and guidelines which place an ecological motivator at the center of development.

The dwellings they have built reflect this. Sharon uses her "junky credits" of manufactured petroleum products because they help her garden on her roof, Alyssa uses an air conditioner because it keeps her straw bale house from rotting, and the Skyhouse apartment complex was built using natural materials alongside sophisticated architectural techniques. Each of these integrations represent a costly endeavor of time, money, and materials. At a time when the typical new construction in the U.S. is trending towards bigger dwellings, the approach at DR has been to prioritize ecological considerations of efficiency and sustainability.

Similarly, Rabbits' resource use and consumption patterns are restrained by their ecological motivator, evidenced both in the covenants and guidelines and in what they actually consume. This is observable in how they fulfill their most basic needs, where the use of permaculture reflects their commitment to sustainable food practices. Or, in how they approach more technology-centered needs in their development of the BEDR electricity microgrid. From solar panels to pooping in a bucket, Rabbits embrace a variety of technologies from ancient to modern in their endeavor to live their ethic of sustainability.

Finally, while there are diverse opinions about technology, all of those are undergirded by the ecological motivator. Some people spend much of their days engaged with digital technologies and readily use new inventions, while others seek to spend as little time in front of a screen or plugging things in as possible. But they all accept that there is a role for new, or "modern," inventions like email in their collective experiment because they enable different parts of DR's sustainability mission.

This has also meant that they have often struggled in disagreement over which type of technology was the most appropriate to deploy. Like much of society, discussions of sustainability at DR are centered around a discourse of the need to shift consumption practices, and the best route to accomplishing that is not always clear. But unique to DR's way of life is a collective understanding that this is not an endeavor to create a *new* and *modern* sustainable era distinct and separate from the old, but one wherein sustainability is a process of development informed by ecological sustainability rather than economic innovation as the primary motivator.

The data in this article is presented from most concrete to most abstract. I have organized it such that what Rabbits have done comes first and what they say they do or what they believe comes second. This is, in part, because actions like building dwellings or growing food are more measurable than expressed beliefs, but also because the ecological motivator requires examples to illustrate. Together, dwellings, resource use, and opinions about technology represent this ethic in different ways, all of which add up to a cohesive project distinct from, and in resistance to, that of the modern society they live in.

This ecological motivator is not present in the same way in other ecovillage experiments. While the literature is replete with examples of ecovillages that have successfully reduced their ecological impact, this is something they have done so in a variety of ways. What has worked for DR may not be applicable for Cloughjordan, Findhorn, Siben Linden, Piracaia or Porongaba as these reside in different cultural contexts. Future research should seek to draw points of comparison and contrast between these experiments, and then use those to inform positive scientific work in the field. By identifying commonalities among successful ecovillages and culturally relevant limitations, we will be able to better identify what works about ecovillages and better predict how these experimental communities can lead us towards more sustainable societies in general. Perhaps some of their innovations can find purchase in the larger society. Or, better yet, perhaps they can demonstrate how the pursuit of progress under the direction of an ecological motivator leads to a worthwhile and fulfilling life while also being more sustainable.

## 6. Conclusions

DR's account of the times incorporates both ancient and modern ideas towards the struggle of sustainability. In Latourian terms, the shovel is prized as much as the laptop if it helps Rabbits live more sustainably. Products and materials deemed "natural" are symbolically weighted as more important than manufactured or "artificial" ones. Yet even as Rabbits build natural houses, they are constructed to contain modern electrical hookups and other manufactured materials that are perceived to make those systems more sustainable in the long run. The integration of different ideas, materials, and technologies also range from simple to complex, with individual Rabbits preferring to either live simply with few digital technologies and others who embraced a wide variety of tools for living sustainably including the internet and portable electricity usage monitors. These integrations are further colored by what is available to individual members based on their socioeconomic status and by the lack of local regulations that allow experimentation in how they build their dwellings. In each of these spectra of ideas and practices, the collective lifestyle at DR is one that embraces a range of options rather than placing emphasis on a more sustainable past where humans were more in touch with nature or a modern and enlightened one that they have engineered beyond nature.

All this is to say that Rabbits are not *anti*-modern—they do not resist much of the progress in technology that modernists would claim distinguishes an era—but that they relate to technology in defiance of the project of modernism. To view the world through a modernist lens would be a hinderance to their mission and in contradiction to their lifestyle. Pooping in a bucket is as innovative and important as using the internet for fundraising and recruitment, since both are key to achieving the mission of the ecovillage and their use is informed by the ecological motivator. Rabbits are perhaps on to something in their approach, too. If the results above from both Lockyer and Jones' audit studies are any indication, this approach to technology has been highly successful in supporting their ethic of sustainability. Rabbits have dramatically reduced their ecological footprint, thanks at least in part to their approaches and attitudes towards technology.

**Funding:** This research received no external funding

**Acknowledgments:** The author is solely responsible for the data gathered and the content of this article.

**Conflicts of Interest:** The author declares no conflict of interest.

### Acronym Guide:

IC          Intentional Community
DR          Dancing Rabbit
CSCC        Center for Sustainable and Cooperative Culture at Dancing Rabbit
BEDR        Better Electricity for Dancing Rabbit, the internal micro-grid electric co-op

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
