# Peer review of "Ecovillagers’ Assessment of Sustainability: Differing Perceptions of Technology as a Differing Account of Modernism"

_sustainability, doi:10.3390/su11216167_

Round 1

Reviewer 1 Report

  This manuscript is based on an ethnographic study of an ecovillage in the United States where people are attempting to live in more ecologically sustainable manners. Using observations from the author's research and the ecovillagers' own published comments, the author examines their uses of an attitudes toward modern technologies. The manuscript reveals an alternative approach to progress and degrowth whereby ecovillagers demonstrate an ambivalent attitude toward modern technologies, using them where they enable them to further their goals and relying on less technologically "advanced" methods and technologies where those are judged a better method of advancing ecological sustainability.

  The manuscript's strengths lie in its subject matter (a rural ecovillage) and its demonstration of ecovillagers' unique attitudes about technology as part of their approach to ecologically sustainable living. Ecovillages are worthy of greater scholarly analysis and public attention. Equally, conclusions that technological "progress" is the path to ecological sustainability are worthy of increased scrutiny and this manuscript achieves that through an analysis of a group of people for whom such scrutiny appears to be a central component of their collective endeavor. While the manuscript in general is worthy of publication, there are several areas that need improvement. First, it is clear that the author rushed to complete the manuscript as there are multiple grammar and syntax errors, missing words, awkward phrasings, run-on sentences, etc. The result of this is that the author's arguments are poorly articulated, unclear, inconsistent and/or unconvincing at several points in the manuscript (see below for specific examples). In addition, as there are significant debates about the validity and reliability of ethnographic methods, the author should more fully describe his data and data collection procedures. (This reviewer is inclined to accept the validity and reliability of ethnographic methods, but nonetheless believes that ethnographers should be specific and strategic in describing their research and data collection). Finally, this reviewer takes issue with at least two of the author's choices of theoretical and/or conceptual terminology. Most significantly, the author's multiple descriptions of ecovillagers' attitudes toward technology as "diachronic" appears to be a misapplication of the term and this reviewer strongly encourages a different choice of terminology to describe their selective adoption of old or new technologies.

  Here are some suggestions for revisions on a line by line basis:

19-21: This characterization of different conceptualizations of sustainability or eco-friendly should be supported with citations.

46, citation 1 and 49, citation 2: Tim Miller's work, especially his 2010 article in Communal Societies provides a more definitive definition of IC.

52, citation 3: Don Pitzer 1997 provides a more definitive account of the history of ICs in the U.S.

64, citation 4: The Gilman's have since updated their definition of ecovillage to include additional dimensions. This updated definition should be cited.

71: The characterization of ecovillages as a form of "retreat" from society is questionable here, especially since the author indicates in the next sentence that he doesn't really mean "retreat". Engagement with Lucy Sargisson's concept of "estrangement" would be more nuanced and productive. See 2007 in the journal Utopian Studies.

90-96: The author uses a community that is not an ecovillage to illustrate the characteristics of an ecovillage from the topic sentence. Choose a different example.

90-110: The argument in these two paragraphs is not well-articulated. I encourage the author to rework this small section.

paragraph starting 145: should be accompanied by a map illustrating the location of the ecovillage

146: "Amish modesty"? The author has not indicated that this is Amish country. Is it? If so, include a short identification of such.

148-149: It is unclear how ecovillagers' neighbors "are a force for constraining some of the more radical ideas they might embrace". Please make it clear how this force expresses itself.

159: Do ecovillagers disseminate "their lifestyle" or their ideas?

160: use of "retreatism"; see above suggestion about engaging with Sargisson's concept of estrangement

175: Why is there a page number but no citation? The author is clearly referring to another published study, but there is no direct quote.

180-182: Here is a direct quote from the same study in 175 above, but there is no page number. All direct quotes from published works should include page numbers.

187-182: The author states that the ecovillagers are "restrained" through their ecological covenants, but these covenants are never described. Please describe the covenants.

189: This is the first instance of the author using a participant's name. In his methods section, he should indicate that he will be doing this later in the paper.

194: The statement that ecovillagers' "degrowth mindset" "dismisses" modernist assumptions about technology contradicts the author's main concluson that they are ambivalent about technology.

200: Don't the ecovillagers have their own neighborhood association rules; is that not what their "ecological covenants" are?

220: It would be useful to know how many houses there are at DR so that statement that "slightly over half" of them are straw bale construction  is more meaningful.

290: This paragraph is about design, not "methods". It is also a clear example of specifically passive design and it should be identified as such.

305: All quotations from interviews should include citations

315: "As shown above..." Do you mean the previously referenced studies by Lockyer and Jones? This article does not clearly demonstrate this point.

337: Here is another reference to membership covenants; these need to be identified and described earlier in the article.

400-401: Does the author have data to support his assertion that only one ecovillager never used the internet?

458-463: This paragraph is completely disjointed and its significance is unclear. Rework.

498-502: This is a run-on sentence and its meaning is unclear.

I have not commented on specific places where apparently rushed writing resulted in significantly awkward phrasing throughout the manuscript and the editors would do well to work with the author on this.

Author Response

I have endeavored to provide a thorough revisiting of the grammatical and phrasing errors, including and beyond what the reviewer took the time to specifically note. Yes, the manuscript was a bit more rushed than I would have preferred, and I appreciate the reviewer’s willingness to endorse its conclusions despite the shortcomings in writing style.

I’ve made several updates to include notable works on ICs not present in the first draft, including works from Sargisson and several from Miller. I’ve also designed a map for inclusion with the paper, per the reviewer’s request. For Robert Gilman’s updated definition – I couldn’t find a primary source on this, only secondary source magazine articles reporting on it. I would gladly update this definition if I could find a reliable source for where he originally expressed it.

Also, I’ve worked to clarify much of what the reviewer found problematic in the definitions and word usage throughout. There is a link in the footnotes to the community’s covenants, but I’ve added some examples. If the reviewer wishes, I could simply include all of them in a table. Some more clarity about how much description of these is needed would be helpful.

Hopefully by extending and editing the literature review, and adding a discussion section my points have been made clearer as well. One point the reviewer notes –  “ 194: The statement that ecovillagers' "degrowth mindset" "dismisses" modernist assumptions about technology contradicts the author's main conclusion that they are ambivalent about technology.” It should be clearer now that while individual Rabbits have differing opinions about personal use of technology that adds up to an ambivalence about any particular deployment, they overall have a strongly shared ethic about the role of technology in meeting their vision.

Reviewer 2 Report

Manuscript is interesting in terms of content, however, some elements of scientific research are missing:

Introduction and literature review. The current state of the research field should be reviewed carefully and key publications cited. Present literature analysis is very limited, new publications shall be cited and scientific novelty proved. Method. Description of the research methodology is not detail enough. Author should describe how many semi-structured ethnographic interviews were performed, what questionnaires used, etc. Discussion is missing. According to journal requirements „Authors should discuss the results and how they can be interpreted in perspective of previous studies and of the working hypotheses. The findings and their implications should be discussed in the broadest context possible and limitations of the work highlighted. Future research directions may also be mentioned“.

Author Response

In this submission, I have worked to update and expand the sections this reviewer has focused on. The literature contains several new pieces of information that are both more recent and more specifically focused on the topic of ecovillages and their approach to sustainability. Additionally, I’ve worked to narrow the terms by eliminating what was kind of a clutter of concepts in service to a more clearly defined hypothesis.

I’ve added substantially to the methods section, including a description of how the ethnographic interviews were done. Finally, I’ve added a discussion section that also contains a section suggesting some directions for future research alongside the shortcoming of the case study at hand.

Reviewer 3 Report

This article promises far more than it delivers. This is largely due to a lack of fit between the empirical content describing practices in Dancing Rabbit (DR) ecovillage and a wider set of claims relating to technology use and concepts such as modernism, degrowth and progress. The central problem of the article is that the empirical data fail to substantiate the wider claims about DR embodying 'an alternative notion of progress' and a 'diachronic conception of the world' (10). These conclusions rest on a discussion of technology use which is interesting but falls far short of the claims made for it. Partly this is due to a failure to robustly define what might constitute 'an alternative notion of progress' or to clearly link the various core concepts used to one another. Thus, for example, the term 'degrowth' is used but there is no discussion of how degrowth relates to progress or to modernism (is degrowth an 'alternative notion of progress'?), nor of how the ecovillage practices described in the article relate to the concept of degrowth. This said, the article does offer promise but would require substantial revision to realise this promise. My recommendation is that the author define much more sharply what is the purpose of the article. As presented in the abstract (though not elaborated anywhere in the text of the article), the author sets up an opposition between 'modernist assumptions' of technology use and 'primitivist notions', seeing DR as offering an alternative to these. The weight of the argument rests on technology use so that other aspects of what would normally be thought of as modernist conceptions of progress (commodified consumption patterns, notions of individual liberty, secularism, to name but a few) are not considered. The author therefore needs to justify this exclusive focus on technology, discussing much more fully the relationship of technology to modernist notions of progress and offering a methodology to guide examination of the empirical study and how it links to the wider theoretical argument being advanced. This will require substantial revisions, much more nuanced consideration of the empirical data and a more fully elaborated and appropriate theoretical framework through which to interpret the empirical evidence.

Author Response

I’ve tried to respond to the critique of delivering less than promised by dialing back on the promise while maintaining the integrity of the delivery. I’ve tried to be more careful about the use of various theoretical ideas throughout. My main goal in this revision has been to reduce the clutter of concepts I had initially tried to relate directly to the data and instead worked to narrow and develop a relationship between the data and a more focused theoretical framing.

This includes a greatly expanded methods section where more work is done to connect the literature to the data, and a discussion section near the end which attempts to do more of the same. Limitations and directions for future research are included in this. I’ve also included some of the major components of the “alternative notion of progress” in the ecological covenants and sustainability guidelines, which I consider more concrete measures of DR’s mission and progress towards it.

As for the focus on technology – I agree that in the context of blind review it seems odd for this to be the sole focus. However, this was submitted for a themed technology and sustainability special issue ("Social Dimensions of Human-Technology Interaction and Sustainable Energy Communities"). So, I can either edit to justify based on the idea that the paper might be read independently from the issue, or leave that alone and let the issue be the context. The other assumptions of a modernist project that the reviewer notes – individualism, secularism, commodification – all do warrant discussion but risk an article that deviates from the theme and becoming too broad and unfocused. That being said, I am open to providing more discussion of modernism to background the discussion of technology.

Round 2

Reviewer 1 Report

I am satisfied with the substantive changes made by the author following the first round of review. Almost all of my comments were addressed and changes made in response to other reviewer comments significantly improved the clarity, argument, and presentation of the manuscript. There are still a number of relatively minor errors in grammar and syntax as well as awkward phrasing that will need to be addressed in the final round of editorial review.

Reviewer 2 Report

The manuscript has been significantly improved and now warrants publication in Sustainability.

Reviewer 3 Report

I have now read the revised version of this article and am impressed by the extent of revisions, the far greater coherence of the empirical data with the theoretical framework that is now achieved